# Changes in the Proteome of Platelets from Patients with Critical Progression of COVID-19

**DOI:** 10.3390/cells12172191

**Published:** 2023-09-01

**Authors:** Monika Wolny, Svitlana Rozanova, Cornelius Knabbe, Kathy Pfeiffer, Katalin Barkovits, Katrin Marcus, Ingvild Birschmann

**Affiliations:** 1Institut für Laboratoriums- und Transfusionsmedizin, Herz- und Diabeteszentrum NRW, Universitätsklinik der Ruhr-Universität Bochum, 32545 Bad Oeynhausen, Germany; 2Medizinisches Proteom-Center, Medical Faculty, Ruhr-University Bochum, 44801 Bochum, Germany; 3Medical Proteome Analysis, Center for Protein Diagnostics (ProDi), Ruhr-University Bochum, 44801 Bochum, Germany

**Keywords:** platelets, proteomics, mass spectrometry, COVID-19, inflammation, acute-phase proteins, NFκB, glucocorticoid receptor, carbonic anhydrase 1, ECMO

## Abstract

Platelets, the smallest cells in human blood, known for their role in primary hemostasis, are also able to interact with pathogens and play a crucial role in the immune response. In severe coronavirus disease 2019 (COVID-19) cases, platelets become overactivated, resulting in the release of granules, exacerbating inflammation and contributing to the cytokine storm. This study aims to further elucidate the role of platelets in COVID-19 progression and to identify predictive biomarkers for disease outcomes. A comparative proteome analysis of highly purified platelets from critically diseased COVID-19 patients with different outcomes (survivors and non-survivors) and age- and sex-matched controls was performed. Platelets from critically diseased COVID-19 patients exhibited significant changes in the levels of proteins associated with protein folding. In addition, a number of proteins with isomerase activity were found to be more highly abundant in patient samples, apparently exerting an influence on platelet activity via the non-genomic properties of the glucocorticoid receptor (GR) and the nuclear factor κ-light-chain-enhancer of activated B cells (NFκB). Moreover, carbonic anhydrase 1 (CA-1) was found to be a candidate biomarker in platelets, showing a significant increase in COVID-19 patients.

## 1. Introduction

Platelets are the smallest cells in human blood, about 3 µm in diameter. The lifespan of a platelet is about 7 to 12 days. These anucleated cells are released into the bloodstream by megakaryocytes and are mainly known for their role in primary hemostasis [1]. Platelets are very sensitive to environmental conditions and are usually activated by lesions in the vessel wall. In addition to the receptors required to maintain hemostasis, platelets possess a number of receptors and surface proteins that are linked to immune system function, such as toll-like receptors (TLRs) or major histocompatibility complex class I. These enable platelets to interact directly with invading pathogens [2]. All this points to the crucial role of platelets in the immune response. For dengue and influenza viruses, for example, it has been shown that these can be internalized by platelets, particularly via TLRs [3,4]. In addition, platelets contain different types of granules that are released upon activation or interaction with pathogens. In addition to growth factors, the alpha granules contain various chemokines and cytokines, which are key players in the regulation of the immune response [5]. The dense granules carry several small molecules, such as adenosine diphosphate, pyrophosphate, and calcium ions, which play a role in the activation of platelets. Enzymes for the degradation of macromolecules, like lipids or proteins, are located within the lysosomes [6,7]. As early as 2002, molecules secreted by platelets such as platelet factor 4 and C-C motif chemokine ligand 5 were associated with an antimicrobial effect [8]. Apart from direct interaction with pathogens, platelets also interact with other cells of the immune system. For example, in the context of thromboinflammation, platelets activate monocytes, macrophages, and neutrophils. The latter capture pathogens by forming neutrophil extracellular traps [9].

In coronavirus disease 2019 (COVID-19), which is caused by severe acute respiratory syndrome coronavirus type 2 (SARS-CoV-2), it is also becoming increasingly clear that hemostasis, particularly platelets, plays a key role. In severe cases of COVID-19, there is an excessive activation of platelets triggered by the virus itself, with hypoxia resulting from lung injury and various components of the immune response, such as cytokines and complement. Moreover, additional stress factors, such as cardiovascular risk factors and predispositions like diabetes, obesity, and old age are often present in patients with a severe course, which, in turn, have an unfavorable influence on platelet activation [10].

Platelet activation leads to the release of platelet granules, which is accompanied by the release of chemokines and cytokines. This presumably exacerbates ongoing inflammation and contributes to the cytokine storm observed in severe COVID-19 cases [11]. COVID-19 patients have high numbers of platelet–neutrophil aggregates [12] and increased formations of neutrophil extracellular traps [13], which can lead to the formation of immunothrombosis and contribute to thromboembolic complications. This eventually leads to platelet consumption and thrombocytopenia, which are already associated with increased mortality [14,15].

Several studies have demonstrated a hyperreactive phenotype in platelets in severe COVID-19 [11,16,17,18,19]. Manne et al. found changes in the transcriptome of such platelets related to protein ubiquitinylation, mitochondrial dysfunction, and antigen presentation [16]. Proteomic studies revealed significant changes in the expression of proteins related to cell death and antiviral response [17], as well as platelet activation and granule secretion [20,21]. In addition, increased levels of protein disulfide isomerases, among others, were found in platelets from COVID-19 patients [22].

COVID-19 is associated with high mortality. In Germany, about 24.5% of all COVID-19 patients admitted to hospital had to be treated in intensive care, and the mortality rate of intensive care patients reached 33% [23]. Given that platelets play an important role in the progression of severe disease, the aim of this study was to further elucidate the role of platelets in disease progression, uncover changes in the cells themselves, and identify promising predictive biomarkers in terms of outcome. To this end, during the second wave of the COVID-19 pandemic, we studied the proteomes of critically diseased COVID-19 patients with lethal outcomes, as well as survivors compared with healthy controls.

## 2. Materials and Methods

### 2.1. Study Population

The study was approved by the ethics committee of the Heart and Diabetes Center North Rhine-Westphalia (HDZ NRW) in Bad Oeynhausen (reg. no. 2019-556). Patients with a critical COVID-19 course, as well as healthy controls (≥18 years of age), were enrolled in this study between 6 November 2020 and 22 February 2021. The real-time polymerase chain reaction of nasopharyngeal swabs was used to test for SARS-CoV-2. Venous blood samples were collected after admission to the intensive care unit (ICU) of the HDZ NRW (median: 2 days after admission). In cases of transferring a patient to a normal ward (NW) of the HDZ NRW, another sample was taken (median: 2 days after transfer). Healthy donors were selected from the in-house blood donation service based on the age and sex of the patients. A prior COVID-19 infection was excluded via the determination of IgG antibodies (SARS-CoV-2 IgG II Quant Assay, Abbott Laboratories, Chicago, IL, USA).

### 2.2. Plasma Preparation

Plasma was obtained from citrate-anticoagulated blood. Blood was collected via a central venous catheter or venipuncture into an S-Monovette^®^ 8.2 mL 9NC (Sarstedt AG & Co. KG, Nuembrecht, Germany). Whole blood was centrifuged twice at 2500× *g* for 15 min at room temperature (rt) without brake, aliquoted, and frozen at −196 °C. Long-term storage was at −80 °C.

### 2.3. Platelet Isolation

Highly purified platelets (HPPs) were isolated via density gradient centrifugation as previously described [24], with some minor modifications. In brief, blood was drawn via a central venous catheter or venipuncture into an S-Monovette^®^ neutral, 7.5 mL (Sarstedt AG & Co. KG, Nuembrecht, Germany), which was filled with 1.5 mL of CCD buffer (100 mM trisodium citrate, 7 mM citric acid, 140 mM glucose, 15 mM EGTA, pH 6.5). Whole blood was centrifuged to obtain platelet-rich plasma (250× *g*, 10 min, brake 0, rt). The platelet-rich plasma was diluted 1:2 with HEPES-NaCl buffer (10 mM HEPES, 145 mM NaCl, pH 7.4), mixed with 20% CCD buffer, and pelleted (1000× *g*, 10 min, brake 0, rt). The pellet was resuspended in HEPES-NaCl buffer. The platelet suspension (5 mL) was separated by density (300× *g*, 20 min, acceleration 1, brake 0, rt) using a density gradient of OptiPrep™ density gradient medium (Sigma Aldrich, St. Louis, MO, USA) and HEPES-NaCl buffer (1.058 g/mL: 15 mL; 1.047 g/mL: 14 mL; 1.037 g/mL 14 mL). Three fractions of the gradient were taken: Fraction 1: 6 mL (plasma proteins), Fraction 2: 11 mL (platelets), Fraction 3: 4 mL (large platelets). Fractions 2 and 3 were pooled if no leukocytes could be detected (determined using Leucocount Human Reagent Kit (BD Biosciences, San Jose, CA, USA) according to the manufacturer’s instructions). In addition, contamination with plasma was excluded by determining the albumin content of Fractions 2 and 3 with the Architect C8000 clinical chemistry analyzer using the MULTIGENT Microalbumin test (Abbott Laboratories, Chicago, IL, USA). Platelets were pelleted (10,000× *g*, 10 min, rt), washed twice with HEPES-NaCl buffer (900 µL), and stored at −80 °C.

### 2.4. Platelets Preparation for MS Analysis

Isolated platelets were resuspended in 50 µL of urea buffer (7 M urea, 2 M Thiourea, 50 mM Tris in distilled water, pH 8.2) containing EDTA-free protease inhibitor cocktail (cOmplete™ Mini, Roche, Basel, Switzerland) and lysed in an ultrasonic bath. The protein concentration was determined in each individual sample according to Bradford [25] using Bio-Rad Protein Assay Dye Reagent (Bio-Rad Laboratories, Hercules, CA, USA). In total, 10 µg of protein was taken for the further preparation steps. Protein disulfide bonds were reduced with 10 mM dithiothreitol for 30 min at room temperature, and free sulfhydryl groups were alkylated with 15 mM iodoacetamide for 30 min at room temperature in the dark. Proteins were precipitated with four sample volumes of pre-chilled 100% acetone at −20 °C overnight. Acetone was removed via centrifugation at 16,000× *g* for 10 min at 4 °C. Protein pellets were washed with 50 µL of pre-chilled 90% acetone, vortexed to mix, and centrifuged at 16,000× *g* for 5 min at 4 °C. Acetone was carefully removed, and the protein pellets were dried for up to 5 min. Pelleted proteins were resuspended in 50 µL of 50 mM ammonium bicarbonate and digested with trypsin (SERVA Electrophoresis, Heidelberg, Germany) at an enzyme-to-substrate ratio of 1:50 at 37 °C overnight. The digestion was stopped with 0.1% trifluoroacetic acid.

### 2.5. Plasma Preparation for MS Analysis

In total, 20 µg of plasma protein (as determined by Bradford assay) was solubilized using urea buffer. Reduction of the proteins was accomplished utilizing 10 mM dithiothreitol, followed by alkylation with 15 mM iodoacetamide. For the digestion process, trypsin (SERVA Electrophoresis, Heidelberg, Germany) was applied at an enzyme-to-substrate ratio of 1:50, and the reaction was carried out at 37 °C overnight. To halt the digestion, 0.1% trifluoroacetic acid was added.

### 2.6. Platelet Preparation for Enzyme-Linked Immunosorbent Assay (ELISA)

In total, 100 µg of platelet pellet was lysed with 800 µL of lysis buffer (150 mM NaCl, 30 mM Tris, 6 mM EDTA, 0.1% NP-40, 1% protease inhibitor P8340 (Sigma Aldrich, St. Louis, MO, USA), 1% phosphatase inhibitor (Cell Signaling Technology, Danvers, MA, USA), pH 7.5) for 30 min on ice and then sonicated (10 × 1 s, 72 W). The lysate was centrifuged (14,000× *g*, 30 min, 4 °C), and the supernatant was stored at −80 °C. To determine the protein concentration of the lysate, the protein was precipitated. For this purpose, equivalent amounts of lysate and ice-cold acetone were mixed and incubated overnight at −20 °C. The protein was then pelleted (14,000× *g*, 30 min, 4 °C) and resuspended in phosphate-buffered saline (Thermo Fisher Scientific, Waltham, MA, USA). Protein concentration was determined with a microBCA assay (Thermo Fisher Scientific, Waltham, MA USA) according to the manufacturer’s instructions.

### 2.7. Label-Free nanoLC-MS/MS with Data-Dependent Acquisition (DDA)

The peptide concentration in the analyzed samples was determined using amino acid analysis following the methods outlined by Plum et al. [26] and May et al. [27]. Based on amino acid analysis, 200 ng of platelet protein underwent analysis through nanoLC-MS/MS, utilizing the previously described procedures [26]. In brief, the extracted peptides were initially injected and pre-concentrated using the Ulti-Mate™ 3000 RSLCnano system (Thermo Fisher Scientific, Waltham, MA USA) with a trap column (Acclaim PepMap 100, 300 μm × 5 mm, C18, 5 μm, 100 Å; flow rate, 30 μL/min). Subsequently, separation of peptides occurred in the analytical column (Acclaim PepMap RSLC, 75 μm × 50 cm, nano Viper, C18, 2 μm, 100 Å) through a gradient ranging from 5% to 30% solvent B over 98 min (solvent A: 0.1% FA in water; solvent B: 0.1% FA, 84% acetonitrile in water; flow rate, 400 nL/min; column oven temperature, 60 °C). The ionization of separated peptides was performed via electrospray ionization, followed by injection into an Orbitrap Fusion™ Lumos™ Tribrid™ Mass Spectrometer (Thermo Fisher Scientific, Waltham, MA USA). During the process, the capillary temperature was maintained at 275 °C, and the spray voltage was set to 1500 V. Internal recalibration employed the lock mass polydimethylcyclosiloxane (445.120 *m/z*). The instrument operated in data-dependent acquisition (DDA) mode with a 2 s cycle time, utilizing higher-energy collisional dissociation (HCD) fragmentation at 28% normalized collision energy (NCE). MS1 measurements were conducted within a mass range of 350–1400 *m/z* with an orbitrap resolution of 120,000 at 200 *m/z* (automatic gain control (AGC) 3 × 10^6^, 80 ms maximum injection time, 1.3 *m/z* wide isolation window, 30 s dynamic exclusion). Fragment analysis took place in an orbitrap mass analyzer with a resolution of 30,000 at 200 *m/z* (AGC 3 × 10^6^, 80 ms maximum injection time).

To evaluate the performance of the nanoLC-MS system, a complex external standard (digest of the human cell line A549) was measured at the start, middle, and end of the study series. Control runs assessed LC pressure profiles, spray stability, symmetry, width, and intensity of chromatographic peaks, as well as retention time stability. Additionally, based on the number of acquired precursor and fragment ion spectra, as well as the number of identified peptides and protein groups, the mass spectrometer’s performance was checked.

The applicability of the generated data for the quantitative analysis was evaluated using the previously described in-house-developed quality control tool MaCProQC [28], implemented into KNIME [29].

The mass spectrometry proteomics data from the DDA analysis were deposited at the ProteomeXchange Consortium via the PRIDE partner repository [30], with the data set identifiers PXD041681 and 10.6019/PXD041681.

### 2.8. Parallel Reaction Monitoring Mass Spectrometry (PRM-MS)—Validation of the Found Changes

A PRM-MS study was designed to investigate whether the selected peptides could reproduce the quantitative differences between the patient and control samples (platelets and plasma) previously reported and determined in this study using DDA. For each of the targeted proteins (carbonic anhydrase 1 (CA-1), serum amyloid A2 (SAA-2), serpin peptidase inhibitor clade A member 3 (Serpin A3), and C-reactive protein (CRP)), at least 2 unique peptides were selected. The uniqueness of the peptides for the “Homo sapience” taxonomy was checked in Skyline (v.22.2.0.255) and using MaCPepDB [31] by comparing them with the UniProtKB database (Homo sapience, reviewed). MS analysis peptides were separated and ionized as described above for the DDA. The instrument was set to acquire both the full MS1 and PRM-MS spectra of the selected peptide precursors (Appendix A). The mass range for the MS1 measurements was set to 350–1500 *m/z* with an orbitrap resolution of 120,000 at 200 *m/z* (AGC 3 × 10^6^; maximum injection time, 200 ms). The instrument was operated in PRM-MS mode with the following parameters: orbitrap resolution was 60,000 at 200 *m/z*, AGC was set to “Standard”, maximum injection time was set to “Auto”, loop control was set to “All”, wide isolation window was 0.5 *m/z*, and HCD fragmentation was at 28% NCE.

### 2.9. ELISA—Validation of the Found Changes

The protein contents of CRP (R&D Systems, Minneapolis, MN, USA; Catalog No.: DY1707), SAA-2 (MyBiosource, San Diego, CA, USA; Catalog No.: MBS2880614), Serpin A3 (RayBiotech, Peachtree Corners, GA, USA; Catalog No.: ELH-SerpinA3), and CA-1 (RayBiotech, Peachtree Corners, GA, USA; Catalog No.: ELH-CA1) were determined in both HPP lysate and plasma via ELISA according to the manufacturer’s instructions. Protein levels were normalized to total protein levels according to a microBCA assay.

### 2.10. Data Analysis

The MaxQuant software (v.2.0.3.1, https://maxquant.org/, accessed on 28 August 2023) was employed for the comprehensive analysis of the DDA raw data. Spectra were meticulously searched against the human reference proteome (UP000005640) from the UniProtKB [32] database (release 2021_11), utilizing specific parameters. The enzyme used for digestion was trypsin, allowing for a maximum of 2 missed cleavages. For peptide tolerance, the values were set to 20 ppm for the first search and 4.5 ppm for the main search, with a fragment match tolerance of 20 ppm and a de novo tolerance of 10 ppm. Methionine oxidation was established as variable modifications, while cysteine carbamidomethylation was considered fixed. To ensure high-confidence peptide spectrum match identification, a reversed decoy-based false discovery rate (FDR) of 0.01 was applied. Protein identification was accomplished, requiring a minimum of 1 peptide and 1 razor peptide. Unique peptides were set at a minimum count of 0 for protein identification, and the protein false discovery rate was set to 0.01. The “match between runs” option was enabled. For quantification, MaxQuant label-free quantification (LFQ) [33] was utilized, with at least one unique peptide at a minimum ratio count of two. The LFQs underwent “classic” normalization [33].

Further analysis involved targeted PRM-MS data analysis using the Skyline software (v.22.2.0.255). For each protein, the most intensive precursor with the five to six most intense “b” and/or “y” fragments was selected for quantification.

The outcomes from both MaxQuant (Appendix A) and Skyline were systematically analyzed and visualized using Perseus 1.6.14.0 [34], R version 4.2.3 (R Core Team, 2021, https://www.R-project.org/, accessed on 28 August 2023), and GraphPad Prism 9 (version 9.0.0). The proteinGroups.txt file from the MaxQuant output was processed in R, and LFQs were employed for further investigations. Missing LFQ values (where LFQ = 0) were suitably addressed by substituting them with NA. The quality of normalization and inter- and intra-group differences were studied using boxplots and principal component analysis (PCA) plots based on normalized log2LFQ intensities, respectively. For PCA plots, only proteins without missing LFQ intensity values were included. To facilitate the reproducibility and sharing of analytical scripts, the R-scripts used in this study are readily accessible at https://github.com/mpc-bioinformatics/QC_Quant (v1_2) (accessed on 28 August 2023).

Pathway and process enrichment analysis for the differently abundant proteins (*t*-test *p*-value < 0.05) was carried out using the web-based METASCAPE database [35] (https://metascape.org, accessed on 28 August 2023). Terms for enrichment analysis included the following ontology categories: COVID, Cell_Type_Signatures, DisGeNET, PaGenBase, TRRUST, Transcription_Factor_Targets, GO Molecular Functions, GO Biological Processes, Reactome Gene Sets, KEGG Pathway, WikiPathways, PANTHER Pathway, Canonical Pathways. All genes in the genome were used as the enrichment background. Terms with a *p*-value < 0.01, a minimum count of 3, and an enrichment factor > 1.5 (the enrichment factor is the ratio between the observed counts and the counts expected by chance) were collected and grouped into clusters based on their membership similarities.

To further capture the relationships between the terms, a subset of enriched terms was selected and rendered as a network plot, where terms with kappa-statistical similarity > 0.3 are connected by edges. The terms were selected with the best *p*-values from each of the clusters, with the constraint that there were no more than 15 terms per cluster and no more than 250 terms in total. The network was visualized using Cytoscape [36], where each node represented an enriched term and was colored by its *p*-value. The STRING database [37] was used for protein–protein interaction analysis.

## 3. Results

### 3.1. Patient Cohort

Patients with a critical COVID-19 course admitted to the ICU of the HDZ NRW were recruited for the study. Within two days after admission, the first blood sample was taken (ICU, *n* = 19). Almost all patients (94.7%) received invasive ventilation, and 86.4% required extracorporeal membrane oxygenation (ECMO). Of the patients enrolled in the study, eleven died (ICU A, *n* = 11), and six survived (ICU B, *n* = 6). For two patients, the outcome could not be clearly identified because they were transferred from the ICU to other hospitals (ICU N, *n* = 2). The survivors were transferred to the normal ward of the HDZ. A follow-up sample could be taken from five patients (NW, *n* = 5). The patients all happened to be male. A 73-country cross-sectional study showed that the case fatality rate was higher in male patients (3.17%) than in female patients (2.26%). In Germany, the difference was lower (2.60% male vs. 2.23% female) [38]. In addition, age- and sex-matched healthy controls with no history of COVID-19 (SARS-CoV-2 IgG II Quant Assay-negative) were recruited from blood donors (*n* = 16) (Table 1, Figure 1a).

### 3.2. Global Proteomic Characterization of Platelets: ICU and Control Platelets Can Be Distinguished Based on Their Proteomes

To further uncover the platelet’s role in COVID-19 development and discover biomarkers for the disease, unfavorable clinical outcome HPPs from 19 ICU patients, 5 NW patients, and 16 healthy controls (Figure 1a) were studied using the LC-MS-based label-free global proteomic approach. Overall, 2017 protein groups (PGs) were identified, showing an overlap of 1828 PGs (hereafter referred to as proteins) for all the group datasets. In contrast to other work [16,39], no viral proteins were detected.

We first analyzed 980 proteins for which at least one missing value was found in the data set. Of these, 603 proteins were identified in at least 50% of the samples in one of the patient groups (Figure 1a). A closer analysis of the proteins revealed 196 proteins with different levels in the patient and control groups (Appendix A, Appendix A). STRING analysis of the proteins showed that they were associated with the acute phase, the inflammatory response, and the antiviral response, as well as the host response to SARS-CoV-1 and SARS-CoV-2. In order to find any specific proteomic features, we first checked which proteins were uniquely identified in a single patient group (found in at least four biological replicates of the corresponding group). The results are presented in Table 2 and Appendix A. Interestingly, most of the proteins found uniquely for the ICU patients were associated with an immune or antiviral response, in particular, an acute-phase response.

To reveal any quantitative differences for the individual patient groups, we next analyzed the levels of 721 proteins quantified with MaxLFQs [33] in every study sample. The PCA based on the protein levels (MaxQuant’s normalized LFQ intensities) in different samples clearly demonstrated distinct clustering for the three sample groups: controls, ICU, and NW (Figure 1b).

Further comparison revealed 25 significantly upregulated and 13 downregulated proteins in ICU compared with the controls (FDR-adjusted *p*-value < 0.01 and fold change ≥ 2) (Figure 2a, Appendix A). A comparison of NW with the controls revealed two proteins (signal transducer and activator of transcription (STAT) 1 and hemoglobin subunit beta) with significantly higher levels in NW (FDR-adjusted *p*-value < 0.01 and fold change ≥ 2) (Figure 2b, Appendix A). Only one protein, FK506-binding protein (FKBP) 5, was found to be significantly different for the ICU vs. NW comparison (higher level in ICU; FDR-adjusted *p*-value < 0.04 and fold change ≥ 1) (Figure 2c, Appendix A). Most interestingly, we were able to find a possible marker to differentiate between non-survivors (ICU A) and survivors (ICU B). STAT1 was significantly more highly regulated in survivors compared with non-survivors (FDR-adjusted *p*-value < 0.04 and fold change ≥ 1) (Figure 2d, Appendix A).

### 3.3. Acute-Phase and Erythrocyte-Associated Proteins in Focus for Biomarker Candidates

Next, we selected proteins uniquely identified in patient platelets (Table 2) and/or the most abundant proteins with missing values, which showed higher intensities in patients compared with the controls (Appendix A), for validation as biomarkers of unfavorable clinical outcomes. Validation was performed using PRM-MS and verified with ELISA. In particular, highly abundant blood acute-phase proteins, i.e., CRP, SAA-2, a member of the SAA family, and Serpin A3, were selectively present in the patient samples. Additionally, SAA-2 was exclusively found in ICU patients. Moreover, these proteins were previously demonstrated to be present at higher levels in the plasma or serum of COVID-19 patients and associated with disease severity [40,41]. Therefore, we further considered these proteins in our validation study.

Regarding CRP, the results for the PRM-MS (Figure 3a–c) and ELISA (Figure 3d–f) measurements showed similar patterns. Platelet lysate from ICU patients presented significantly elevated CRP levels compared with healthy controls (PRM-MS: Figure 3a, ELISA: Figure 3d). In addition, the surviving ICU patients (ICU B) tended to have slightly lower CRP levels compared with deceased ICU patients (ICU A) (PRM-MS: Figure 3a, ELISA: Figure 3d). In turn, the NW patients tended to have lower CRP abundance than the ICU patients (PRM-MS: Figure 3a, ELISA: Figure 3d). A similar pattern was observed for CRP in plasma (PRM-MS: Figure 3b, ELISA: Figure 3e). Furthermore, platelet and plasma levels of CRP showed a high correlation (PRM-MS: Figure 3c; ELISA: Figure 3f).

Both PRM-MS and ELISA studies of Serpin A3 in platelets revealed similar results to CRP (Figure A1a (PRM-MS); Figure A1d,e (ELISA)). ICU patients presented the highest Serpin A3 levels. ICU A and ICU B did not differ significantly but presented a significant increase compared with the controls. NW tended to show a slightly higher level of Serpin A3 than the controls. However, no significant differences were observed with PRM-MS in plasma (Figure A1b), probably because of the higher sensitivity of ELISA compared with PRM-MS. A high correlation was observed between platelet and plasma concentrations via ELISA measurement (Figure A1f).

Since SAA-2 and SAA-1 are closely related proteins with high sequence homology, differentiation with PRM-MS is possible given the higher specificity, and therefore, it probably shows different results than ELISA, which presumably cannot differentiate between the two isoforms. Therefore, we focused on PRM-MS results (Figure A2). The highest SAA-2 amount was found in the ICU patients, both in platelet lysate and in plasma (Figure A2a,b). SAA-2 was detected in platelet lysate exclusively in the ICU patients. Survivors (ICU B) tended to have slightly lower SAA-2 levels. Plasma levels of SAA-2 in ICU patients were significantly increased compared with the controls. Levels in NW patients appeared to be increased compared with the controls. The correlation analysis revealed a high correlation between platelet and plasma levels (Figure A2c).

Additionally, CA-1, which is involved in respiration and the transport of carbon dioxide and bicarbonate between metabolic tissues and lungs [42], was assessed for its biomarker potential, as it was found uniquely in the ICU and NW samples in the global proteome analysis. Furthermore, a significantly increased CA-1 concentration was also found in the plasma of patients severely affected by COVID-19 [43,44]. With PRM-MS, CA-1 could only be found in platelets (Figure 4a). Significantly increased levels were detected in ICU patients ((A+B+N) and ICU A), as well as NW patients, compared with the controls. Survivors (ICU B) tended to present lower CA-1 levels than non-survivors (ICU A), while the CA-1 levels of NW patients tended to be higher than those of survivors (ICU B). A similar picture was also observed with ELISA (Figure 4b). Likewise, there was a tendency toward increased concentration in the NW patients compared with the survivors (ICU B) and controls, the latter being found to be significant. Plasma CA-1 could only be determined with ELISA without any significant differences between the cohorts (Figure 4c). There was no correlation between platelet and plasma concentrations (Figure 4d).

### 3.4. Insights into the Pathogenesis of COVID-19 at the Platelet Level

To characterize changes in the platelets during COVID-19 development, we further analyzed the signature of exclusively expressed and differentially expressed (*p*-value < 0.05) proteins between the ICU and control groups. Pathway and process enrichment analyses of the exclusively expressed proteins showed their high association with COVID-19 and SARS-CoV-2 signaling, as well as with immune response features such as tumor necrosis factor production and complement activation. In addition, an association with platelet degranulation was also found (Figure 5a, Appendix A). An analysis of differentially regulated proteins, as expected, revealed differences in hemostasis and related them to platelet activation and aggregation, as well as changes in cell adhesion regulation. In addition, changes were found in the immune response, associated with cytokine signaling and neutrophil degranulation. Alterations in protein kinase binding, cell adhesion molecule binding, vascular endothelial growth factor, and platelet-derived growth factor signaling were also observed. Moreover, differences in protein folding, protein localization, protein complex assembly, cytoskeletal reorganization, and vesicle-mediated transport were identified (Figure 5b, Appendix A).

An analysis of 186 differentially abundant proteins (*p* < 0.05) for the NW comparison with the control group showed their greatest association with protein folding, along with platelet activation, aggregation, and signaling and cell adhesion molecule binding. Similar enrichment of signaling pathways and processes was found for comparisons of the whole ICU group (A+B+N) or the survivors (ICU B) with the NW group (122 and 116 differently expressed proteins, respectively) (Figure A3, Appendix A). An analysis of the deceased and surviving ICU patients mainly showed a correlation with hemostasis, metabolic reprogramming, and cell adhesion (Figure 6, Appendix A).

The proteins found to be related to protein folding were further examined in more detail (Figure A4). The abundance of these proteins rose with the severity of disease status (Table A1). Only one protein, protein disulfide-isomerase (PDI) A5, showed opposite regulation. Notably, significant downregulation in PDIA5 was observed in the ICU group compared with the control group. Furthermore, FK506-binding proteins (FKBPs), which possess prolyl isomerase activity and act as chaperones in protein folding, could be found in this context. FKBP5 is found among the significantly regulated proteins (Figure 2). The closely related FKBP4 showed exclusive expression in the patient samples (ICU A+B+N and NW) and was not found in the control group (Table 2).

## 4. Discussion

In addition to hemostasis, platelets fulfill a crucial function in the immune response. It is already known that a direct interaction with various immune cells and pathogens occurs [2]. Analyses of the platelet proteome have been performed for various diseases of bacterial and viral origin and have revealed specific changes in platelet protein compositions [45,46]. Alterations in the platelet proteome have also been observed in SARS-CoV-2 infections [20,21,22]. Platelet involvement in COVID-19 seems to play a role in the most affected patients. In a previous study, we found that severely ill patients requiring ECMO respiratory support presented with pronounced thrombocytopenia throughout the course of the disease. This is accompanied by a significantly increased proportion of immature platelets [15]. To reveal the characteristic changes in platelets of an equivalent collective, we performed proteomic analysis. Our work focused particularly on critically ill patients supported with ECMO compared with survivors in this group on the way to recovery and healthy controls.

### 4.1. Potential Predictive Biomarkers for the Outcome of COVID-19 in Patients’ Platelets

In the global proteome analysis, we found several plasma proteins that were exclusively expressed in the patient groups. These exclusively expressed proteins seemed promising in the search for a biomarker for severe COVID-19 disease. In particular, the acute-phase proteins CRP, SAA-2, and Serpin A3 and the respiratory-related erythrocyte protein CA-1 stood out as candidates.

The three acute-phase proteins have already been associated with a predictive function for COVID-19 in various plasma proteome studies [40,41]. CRP and SAA are plasma proteins that, to the best of our knowledge, have only been detected in platelets in a proteomic analysis of platelets from COVID-19 platelets [21] but not in any other context. So far, there is no evidence of endocytosis in CRP or SAA or of increased transcription in the situation of COVID-19 [16]. This suggests that the increased SAA and CRP concentration in platelets is due to the surface binding of the protein. This suspicion is supported by the clear correlation between plasma and platelet concentration. Whether this is a process of endocytosis needs to be further investigated. However, interactions between platelets and CRP or SAA have been described. SAA inhibits aggregation and interacts with platelets via an interaction with the glycoprotein (GP)IIbIIIa receptor [47,48]. On the surface of the platelets, pentameric CRP (pCRP) present in the plasma is converted into monomeric CRP (mCRP), which, in contrast to pCRP, has a proinflammatory and prothrombotic effect [49]. The FcγRIIa (CD32) PAF receptor and phosphocholines have been discussed regarding the binding of pCRP to platelets. mCRP, on the other hand, appears to interact with platelets via GPIb-IX-V and to have an activating effect [50].

Serpin A3, in contrast, is a protein of platelet alpha granules. The main target is cathepsin G, which is predominantly secreted by neutrophils at the site of inflammation and has a proinflammatory effect. Cathepsin G also activates platelets [51]. Serpin A3 inhibits cathepsin G; thus, it has an anti-inflammatory effect, as well as an indirect inhibitory effect, on platelets [52]. This may suggest that the increased concentration of Serpin A3 in platelets is due to increased Serpin A3 translation in megakaryocytes or increased synthesis within platelets as an anti-inflammatory response to severe COVID-19 disease. The latter was supported by the pathway and process analysis of the differentially abundant proteins in the different patient groups. The proteins found exhibited a highly significant association with protein folding, which is discussed in detail below.

CA-1 takes an outstanding position, as we found it significantly increased in the platelets of ICU patients. In contrast to other groups [43,44], we could not find CA-1 in the plasma of the different patient groups or only to a small extent; therefore, there is no correlation between platelet and plasma concentrations. The family of CAs occurs ubiquitously in human cells and is involved in respiration and the exchange of carbon dioxide and bicarbonate with the lungs, among other processes [42]. Severe cases of COVID-19 are characterized by lung injury with the need for mechanical ventilation and support with ECMO. Nearly all ICU patients in this study received invasive ventilation (94.7%), and 68% of these patients were treated with ECMO. The research on carbonic anhydrases in relation to platelets is mainly related to isoform II of the enzyme and is poorly understood. Proton efflux in response to thrombin stimulation and CO_2_ hydration in platelets has been described [53,54]. Current studies show the influence of CAs on the procoagulant response of platelets. Therefore, corresponding inhibitors are also discussed as a new method of inhibiting platelet activity [55].

### 4.2. Cytokine Signaling and Non-Genomic Action of Transcription Factors in Platelets from Severely Affected COVID-19 Patients

Among the most significantly regulated proteins, especially between the ICU A (non-survivors) and ICU B (survivors) groups, was the transcription factor STAT1. This protein is part of the Janus kinase/STAT signaling downstream of interferon receptors. This suggests an antiviral response and reaction to the cytokine IFN-γ. This association was also found in platelets from COVID-19 patients in a previous work [20]. In addition, we found other cell-death-associated proteins (CD274, HTATIP2, and S100A8) in the patients, which supports the observation that severely diseased COVID-19 patients develop thrombocytopenia [15]. Upon ICU sampling, the median platelet count of our cohort was 164 × 10^6^/mL, and the percentage of immature, newly formed platelets was 6.1%. These values are at the lower and upper limits of the normal ranges, respectively. STAT1 is also active in megakaryocyte differentiation [56]. This suggests that it may be a compensatory effect of the decreased platelet count. In addition, thrombopoietin, which activates platelet production, activates various STATs, including STAT1, in human platelets [57]. On the other hand, a previous proteomic analysis of platelets from COVID-19 patients found a reduction in the thrombopoietin receptor [21].

Exclusively expressed proteins in the patient group (ICU (A+B+N) vs. control) showed a clear association with SARS-CoV-2 infection or COVID-19. We could not provide evidence for the presence of SARS-CoV-2-specific proteins, but other groups had SARS-CoV-2 RNA in platelets [16,20]. Including the regulated proteins, it became clear that there were mainly hemostasis-associated changes. This is in line with the literature reporting that platelets from COVID-19 patients have a hyperreactive phenotype [11,16,17,18,19].

Interestingly, several different processes were found that are mainly related to protein localization and folding. In particular, protein folding is of exceptional importance since this process was also found in the comparison with the group of recovered patients (NW vs. control and vs. ICU (A+B+N) and ICU B). It has long been assumed that platelets are unable to generate proteins, as these are anuclear cells and thus do not possess chromosomal DNA [1]. However, platelets use mitochondrial DNA, pre-mRNA, and microRNA processing, as well as mRNA translation, to synthesize proteins [58,59]. This suggests that the platelets of both the ICU and NW groups engaged in increased protein synthesis.

Among the proteins involved in protein folding in platelets, three isomerases stand out: one PDI and two FKBPs. A previous study also found PDIs, particularly P4HB and PDIA6, in COVID-19 platelets [22]. In our patient collective, an exclusive expression of FKBP4, as well as an increased expression of FKBP5, could be found in the patients. FKPB4 and FKBP5 are involved in glucocorticoid receptor (GR) signaling. The GR is present in the cytosol complexed with various proteins, including HSP90 and members of the FKBP family, and translocated to the nucleus upon ligand binding. In platelets, GR must be involved in non-genomic functions, such as platelet activation. It has been shown that platelet aggregation is reduced with adenosine diphosphate and thromboxane A_2_ mimetic U46619 under the influence of prednisolone, a member of the glucocorticoid family. However, this influence is probably ligand-dependent since dexamethasone, another member of the glucocorticoid family, showed no effect on platelet aggregation [60]. In addition, interaction with monocytes, adhesion and spreading on collagen, and shear stress-dependent thrombus formation are inhibited by prednisolone as well [61]. There is additional evidence that GR also plays a non-genomic role in posttranscriptional gene regulation. For example, in arterial smooth muscle cells, it has been shown that the instability of monocyte chemoattractant protein-1 mRNA is caused by dexamethasone-activated GR [62].

A common treatment for severely diseased COVID-19 patients is the administration of dexamethasone, as recommended by both the European Medicines Agency [63] and the National Institutes of Health [64]. Our ICU patient population was also treated with dexamethasone (58% of the ICU patients). In peripheral blood mononuclear cells, FKBP5 expression has been shown to be induced by dexamethasone. Dexamethasone also appears to have an effect on platelet formation. Recovery in immune thrombocytopenia was better with high-dose dexamethasone than with standard-dose prednisolone [65]. Gobbi et al. studied platelet gene expression in acute myocardial infarction. Here, FKBP5 was found to be one of a set of five proteins for the identification of myocardial infarction [66]. FKBP4 is involved in the dynamics of microtubule formation. A direct interaction with tubulin inhibits microtubule formation [67]. This plays a crucial role in shape retention in quiescence, as well as shape change in activated platelets [68]. Our pathway and process analysis showed that there were significant changes in platelet activation and aggregation in the ICU patients.

The family of FKBPs also plays a role in the immunoregulatory nuclear factor κ-light-chain-enhancer of activated B cell (NFκB) signaling, which consists of several subcomponents. An interaction between FKBP5 and IKKα, IKKε, TAK1, and MEKK1 has been demonstrated [69]. Erlejman et al. showed that FKBP4 and FKBP5 have inhibitory and activating effects on NFκB activity, respectively [70]. In platelets, NFκB signaling plays a non-genomic role, is critical for platelet survival, and is associated with the regulation of stress response and apoptosis [71]. In addition, NFκB is involved in platelet activation and aggregation with various agonists, especially thrombin and collagen [72,73]. Platelets also possess TLRs that allow for direct interaction with pathogens. It is well established that TLRs activate NFκB signaling in nuclear cells. This has also been shown for TLR2 and TLR4 in platelets [74].

Our study has a few limitations. Firstly, the critically diseased patients were given a cocktail of different drugs, which were reduced when the patients improved. Thus, the idea that these drugs have an influence on the proteome of the platelets cannot be excluded. Secondly, especially given that platelets only have limited de novo protein synthesis, a proteome study, which mainly investigated differences in protein modification, e.g., phosphorylation instead of protein levels, would be interesting to assess the activation of different signaling pathways more precisely. Furthermore, the transfer of proteins through the cargo of extracellular vesicles would be an interesting aspect for further investigations since it has been shown that platelets can internalize extracellular vesicles [75]. In addition, changes in the proteome of the extracellular vesicles of COVID-19 were also observed; in particular, the proteins CRP and Serpin A3 were also found to be increased in the patient group [76].

## 5. Conclusions

In conclusion, we showed in our study that platelets from critically diseased COVID-19 patients have a specific proteome profile compared with healthy controls and surviving, improved patients. In particular, proteins associated with protein folding are prominent. This suggests a specific de novo protein synthesis in the platelets. In addition, we found a number of proteins with isomerase activity in the patient samples, which seem to exert an influence on platelet activity through the non-genomic properties of GR and NFκB in particular. This hypothesis needs to be investigated in further studies. Exclusively expressed proteins in the patient population also indicate changes in platelet activity associated with both activating and inhibitory effects. Some of the potential biomarkers identified in the plasma/serum of COVID-19 patients were also found in elevated levels within or on the surface of platelets. These platelet-associated proteins showed a strong correlation with their concentrations in plasma, although some slight variations were observed depending on the analytical method used. Among the proteins investigated, CA-1 emerged as the most compelling candidate identified in platelets. Notably, its levels were significantly elevated in deceased ICU patients (ICU A), and this elevation did not exhibit a correlation with its plasma concentration. In contrast, survivors did not display a substantial increase in CA-1 levels in platelets until a later stage (NW samples). We would, therefore, consider this protein to be a candidate for further biomarker identification studies. Furthermore, our results suggest that there must be specific signaling in acute-phase proteins in platelets, which needs to be investigated in detail in the future.

## Figures and Tables

**Figure 1 cells-12-02191-f001:**
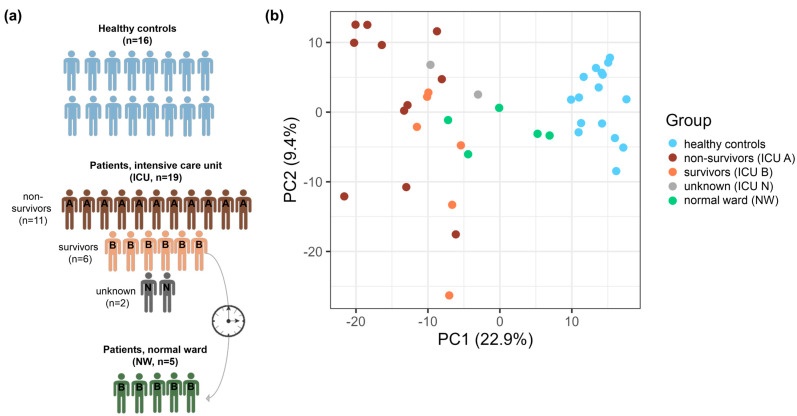
Studied patient cohorts (**a**) and comparison of their proteome profiles using principal component analysis (PCA) (**b**). (**a**) Between November and December 2021, COVID-19 patients (n = 19) admitted to the HDZ NRW in the ICU were included in this study. The highly purified platelets were prepared within two days after patient admission. The majority of patients died (ICU A, n = 11). Two patients had no identifiable outcome, as they were transferred to other hospitals (ICU N). Six patients survived and were transferred to the normal ward (ICU B). The platelets of five of these patients could be prepared at this time point (NW). The control group included age- and sex-equivalent healthy blood donors. (**b**) PCA, based on 721 PGs quantified with LFQs in all the samples, showed a distinct clustering of ICU patients (ICU A, non-survivors: red; ICU B, survivors: orange; ICU N, unknowns: gray), NW patients (green), and healthy controls (blue).

**Figure 2 cells-12-02191-f002:**
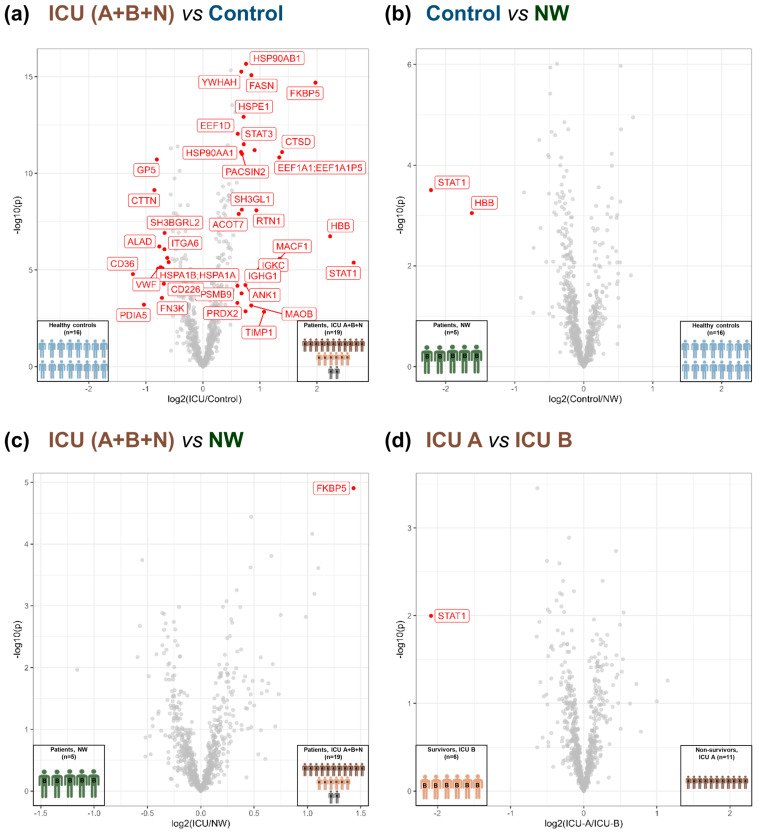
Volcano plot of the different group comparisons. (**a**) ICU (A+B+N) vs. controls based on 723 PGs found in each sample. Of these, 38 were regulated significantly (FDR < 0.01, S0 = 2). (**b**) Control vs. NW based on 771 PGs found in each sample. Among these, two (STAT1 and HBB) were significantly different (FDR < 0.01, S0 = 2). (**c**) ICU (A+B+N) vs. NW based on 784 PGs found in each sample; one protein (FKBP5) was significantly different (FDR < 0.04, S0 = 1). (**d**) ICU A vs. ICU B based on 784 PGs found in each sample; one protein (STAT1) was significantly varied (FDR < 0.04, S0 = 1). ICU: intensive care unit (ICU A: non-survivors, ICU B: survivors, ICU N: unknown); NW: normal ward.

**Figure 3 cells-12-02191-f003:**
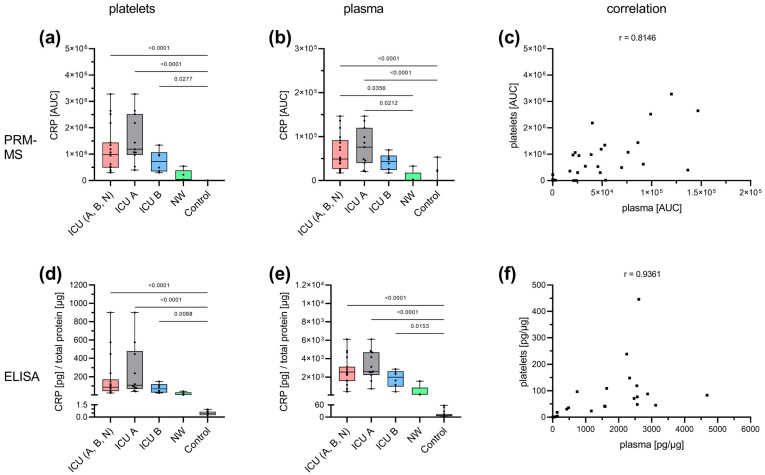
Specific analysis of CRP. CRP was analyzed with PRM-MS (**top row**) and ELISA (**bottom row**) in both platelets (**a**,**d**) and plasma (**b**,**e**). A similar pattern was observed with both methods. ICU patients showed the highest CRP level, which was significantly increased compared with the control group. Survivors (ICU B) tended to have marginally lower CRP levels than non-survivors (ICU A). NW tended to present lower CRP levels than ICU. A similar pattern was observed in plasma samples. A correlation between platelet and plasma levels could be demonstrated by both PRM-MS (*p* < 0.0001) (**c**) and ELISA (*p* < 0.0001) (**f**). Platelet and plasma readings are presented as boxplots. Data were analyzed using the Kruskal–Wallis test followed by Dunn’s multiple comparison. The correlation was determined according to Spearman. ICU: intensive care unit (ICU A: non-survivors, ICU B: survivors, ICU N: unknown); NW: normal ward.

**Figure 4 cells-12-02191-f004:**
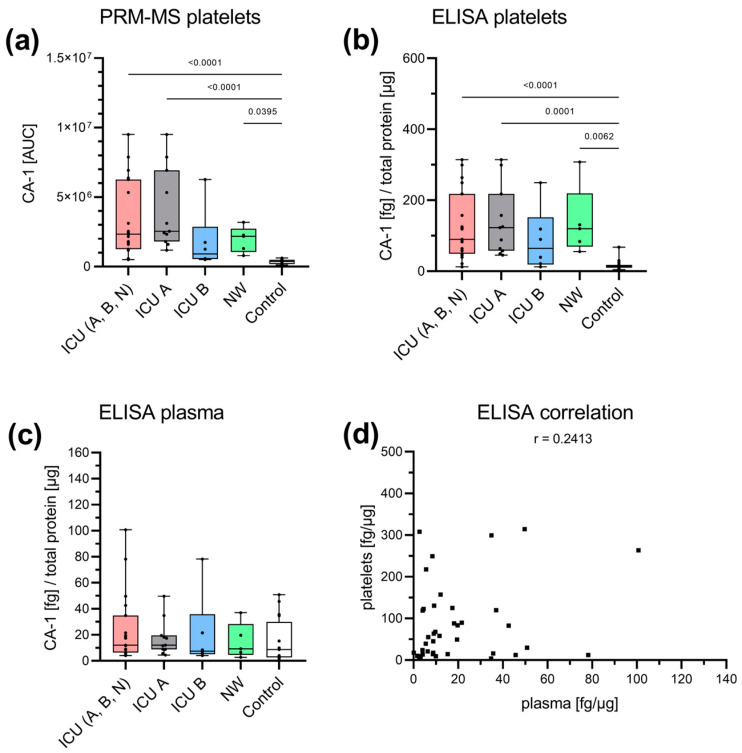
Specific analysis of CA-1. CA-1 was analyzed with PRM-MS and ELISA in both platelets (**a**,**b**) and plasma (**c**). With PRM-MS, CA-1 could only be detected in the platelet lysate. The abundance was significantly higher in the ICU group ((A+B+N) and ICU A), as well as the NW group, than in the control group. There was a tendency toward a decrease in ICU B (survivors) compared with ICU A (non-survivors) and NW. A similar pattern was found with ELISA. No changes were detected in plasma using ELISA. There was no correlation between platelet and plasma concentrations (*p* = 0.1336) (**d**). Platelet and plasma readings are presented as boxplots. Data were analyzed using the Kruskal–Wallis test followed by Dunn’s multiple comparison. The correlation was determined according to Spearman. ICU: intensive care unit (ICU A: non-survivors, ICU B: survivors, ICU N: unknown); NW: normal ward.

**Figure 5 cells-12-02191-f005:**
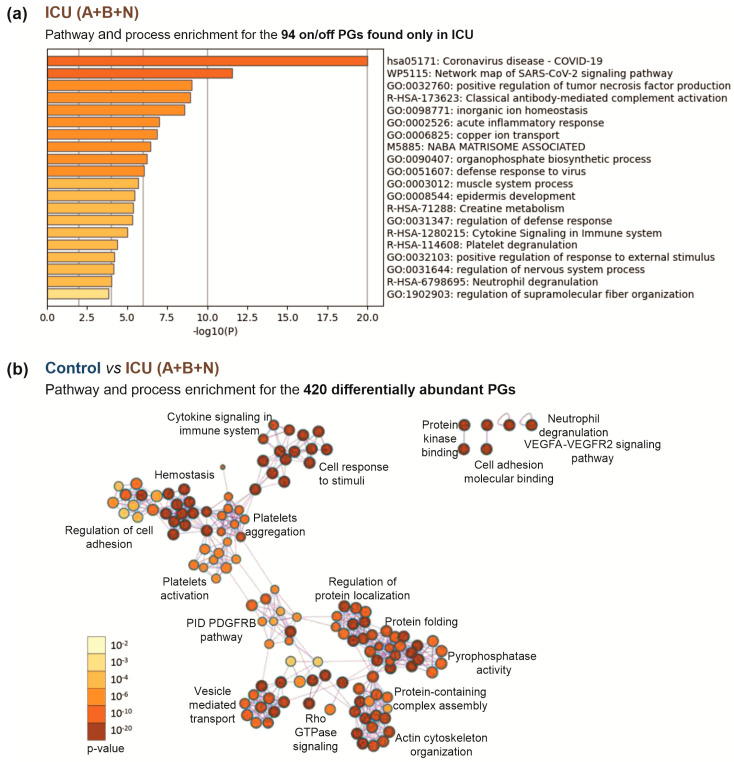
Comparison of the ICU patient group (A+B+N) with the healthy controls. (**a**) The top 20 enriched biological processes across 94 PGs exclusively found in ICU. Each bar represents a statistically ranked enrichment term. The proteins selectively present in the patient group showed associations with SARS-CoV-2 and COVID-19 and the further involvement of the immune response in pathway and process enrichment analysis. (**b**) Network of enriched terms for the 420 differentially abundant proteins in controls and ICU patients (*p* < 0.05) colored by *p*-value. Terms containing more genes tend to have a more significant *p*-value; the darker the color, the more statistically significant the node (see legend for *p*-value ranges). Pathway and process analysis of regulated and exclusively expressed proteins showed correlation with various mechanisms, especially hemostasis, platelet aggregation and activation, and protein folding and localization. ICU: intensive care unit (ICU A: non-survivors, ICU B: survivors, ICU N: unknown); NW: normal ward.

**Figure 6 cells-12-02191-f006:**
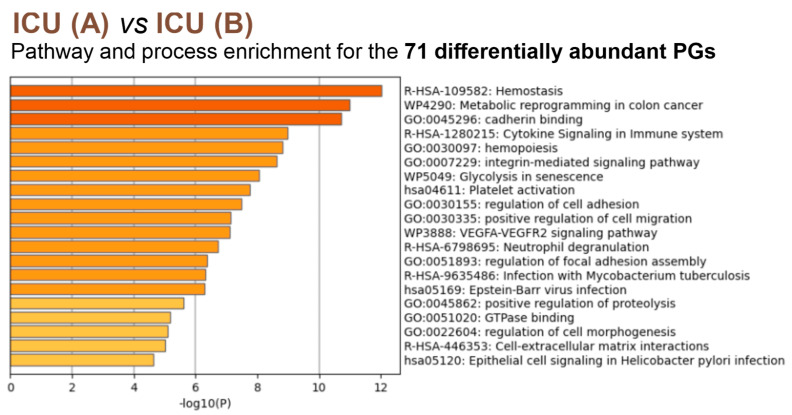
The top 20 enriched biological processes across 71 differently abundant PGs (*p* < 0.05) from the comparison of ICU A vs. ICU B, colored by *p*-values (the darker the color, the more statistically significant). Each bar represents a statistically ranked enrichment term. Pathway and process analysis showed a clear correlation with hemostasis. In addition, changes in metabolism and cell adhesion were observed. ICU: intensive care unit (ICU A: non-survivors, ICU B: survivors, ICU N: unknown); NW: normal ward.

**Table 1 cells-12-02191-t001:** Patients’ baseline characteristics. COPD: chronic obstructive pulmonary disease; CRP: C-reactive protein; ECMO: extracorporeal membrane oxygenation; ICU: intensive care unit (ICU A: non-survivors, ICU B: survivors, ICU N: unknown); IL-6: interleukin 6; INR: international normalized ratio; NW: normal ward; PTT: partial thromboplastin time.

	ICU (A+B+N)	NW	Control
n	19	5	16
age, median (range) [a]	62 (46–81)	61 (46–69)	61.5 (48–70)
male, n (%)	19 (100.0)	5 (100.0)	16 (100.0)
Platelet count on sampling, median (range) (10^9^/L)	164 (25–309)	243 (213–253)	285.5 (179–414)
immature platelet fraction on sampling, median (range) (%)	6.1 (2.7–11.5)	2.0 (0.9–4.7)	3.4 (1.6–14.3)
**Hospitalization**			
total stay, median (range) (d)	30 (4–113)	51 (35–78)	
ICU, median (range) (d)	24 (4–113)	34 (24–63)	NA
NW, median (range) (d)	12.5 (6–26)	15 (6–26)	
**Outcome at discharge**			
deceased, n (%)	11 (57.9)	0 (0.0)	
improved, n(%)	6 (31.6)	5 (100.0)	NA
unknown, n(%)	2 (10.5)	0 (0.0)	
**Comorbidities**			
adipositas, n (%)	7 (36.8)	2 (40.0)	
hypertension, n (%)	12 (63.2)	3 (60.0)	
hyperlipoproteinemia, n (%)	4 (21.1)	1 (20.0)	
diabetes mellitus type II, n (%)	7 (36.8)	2 (40.0)	NA
cardiac disease, n (%)	11 (57.9)	3 (60.0)	
COPD, n (%)	2 (10.5)	0 (0.0)	
renal insufficiency, n (%)	5 (26.3)	1 (20.0)	
**Treatment**			
invasive mechanical ventilation, n (%)	18 (94.7)	5 (100.0)	
ECMO, n (%)	13 (68.4)	2 (40.0)	
ECMO duration, mean (d)	17.5	9.4	
convalescent plasma, n (%)	7 (36.8)	2 (40.0)	NA
platelet concentrate, n (%)	14 (73.7)	1 (20.0)	
platelet concentrate number, mean	4.5	0.4	
dexamethasone, n (%)	11 (57.9)	0 (0.0)	
**Laboratory findings on admission**			
leukocytes, median (range) (10^9^/L)	12.3 (4.1–46.0)	10.1 (4.1–18.9)	
erythrocytes, median (range) (10^12^/L)	3.6 (3.0–5.0)	4.34 (3.73–5.04)	
platelets, median (range) (10^9^/L)	210 (73–770)	188 (126–331)	
CRP, median (range) (mg/dL)	19.0 (4.7–53.0)	14.0 (9.2–26.0)	
IL-6, median (range) (pg/mL)	116 (26–938)	88 (46–143)	NA
fibrinogen, median (range) (mg/dL)	624 (286–900)	543 (286–624)	
d-dimer, median (range) (mg/L)	3.5 (0.6–35.2)	2.3 (0.6–35.2)	
PTT, median (range) (s)	29 (23–200)	25 (23–46)	
INR, median (range)	1.1 (1.0–1.9)	1 (1.0–1.9)	

**Table 2 cells-12-02191-t002:** Uniquely identified proteins found in at least 4 samples of the corresponding group. ICU: intensive care unit (ICU A: non-survivors, ICU B: survivors, ICU N: unknown); NW: normal ward.

Patient Group	Relevant Processes	Protein (Gene Name)
Control, NW	Glycogen metabolism	PPP1R3E
	Signaling, trafficking	CD63
Control, NW, ICU B	Signaling, trafficking, cell cycle regulation	PAPSS2, RGS7, CDK3, RAB30, PLEKHF2
	Focal adhesion	TNS1
ICU A+B	Acute phase	SAA1, SAA2, ORM1
	Antiviral response	OAS2, OAS3, PARP9, PIK3R1, DDX58
	Ribosomal proteins	RPL12, RPS14, RPS4X, RPS3A
	Apoptosis and inflammation	PYCARD, STAT2
	Nucleotide synthesis	CMPK2
	Metabolism	PPT1, TFRC, BCKDK, ATP6V1D, GATM
	Binding activity regulation	OLFML2A, LGAL
ICU A+B, NW	Acute phase, defense response	CRP, APOD, LGALS3BP, ITIH3, IGHG3, IGHM, IGKV3D-20
	Protein folding	FKBP4, CCDC47
	Ribosomal proteins	RPS3, RPS7, RPL18, RPS18, RPS28
	Cell death	CD274, HTATIP2, S100A8
	Erythrocyte	CA1
	Nucleic acid binding	YBX1, UBAP2L, SERBP1, PRKAR1B
	Metabolism	OPLAH, CP
	Other	ECHDC1, MAP1B, PITPNM2

## Data Availability

The mass spectrometry proteomics data from the DDA and PRM analyses were deposited at the ProteomeXchange Consortium via the PRIDE partner repository with the data set identifiers PXD041681 and 10.6019/PXD041681.

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
