# Peer review of "Changes in the Proteome of Platelets from Patients with Critical Progression of COVID-19"

_cells, 2023, doi:10.3390/cells12172191_

Round 1

Reviewer 1 Report

Wolny and colleagues investigate alterations in the platelet proteome between patients admitted to the ICU following COVID-19 complications and healthy controls. While this is a potentially interesting set of observations, my major and minor concerns are itemized below.

Major points

Conceptual- The major conceptual messages of this manuscript are that a) expression of protein folding proteins is differentially abundant between healthy controls and patient’s with COVID-19, and b) carbonic anhydrase I (CA-1) protein expression from a purified platelet lysate appears to be correlated in patients with COVID-19. The author’s chose to validate the MS/MS data using ELISA assays. ELISA or western blotting that detect FKBP4 would strengthen this manuscript by showing a validation of the MS/MS studies supporting (a). Furthermore, the study participants were exclusively male, though the authors negate to address this entirely. Retrospective studies have previously established a difference in mortality and propensity to experience severe COVID-19. This should be clear and directly addressed within the manuscript.

Technical-

1-      Figure 1b- Regarding the PCA analysis, it is unclear what variables were used for the clustering and how they were weighted to arrive at this final graph. Are the raw data going to be added as supplemental data or deposited prior to publication? Perhaps an isolated hierarchical heat map would also be helpful, rather than exclusively a list of uniquely identified proteins (as depicted in table 2).   

2-      Can the authors clarify the sequences used for the PRM analysis? ESISVSSEQLAQFR (light), EQLSLLDR (light), EIGELYLPK (light), ADLSGITGAR (light), NLAVSQVVHK (light), ITLLSALVETR (light) are not unambiguous sequences. If the MS/MS was done label free, how can the authors confirm these sequences are definitely CA-1 (not CA-2) or Serpin a3 (not multiple other targets) respectively?

3-      ELISA validation. This reviewer cannot easily locate all of the listed ELISA kits listed in the methods section. The specific human serum amyloid A2 material used from mybiosource is unclear, in addition to the carbonic anhydrase I from RayBiotech. Can the authors provide catalog numbers to clarify this point?

4-      Data depicted in Figure A2 is quite unclear. According to table 2 both SAA1 and SAA2 were uniquely identified in ICU patients, not in healthy controls. The authors note on lines 370-371 that the ELISA data are not concordant with observations from PRM. However, this reviewer does not understand the explanation given, namely that the ELISA specificity between SAA1 and SAA2 may be the confounding factor. According to this reviewer, a) if the authors think this data cannot be interpreted due to technical issues, it should be removed prior to publication, and b) as stated in another comment, the authors should clarify the starting material and expected range of SAA2 protein for the ELISA, or consider other options including western blotting.

5-      There appears to be a disconnect on the number of protein groups analyzed in Figures 5, 6, and A3. (Line 428 says 168 proteins were assessed for Figure A3a, and there are 186 in the excel file; figure 5a data unclear if it is located in excel; Figure 6 legend states 72 PGs were assessed, but 71 are listed in the excel file; similar concerns with other figure A3b-c).

Minor points

1-      Not all the acronyms are spelled out the first time. A non-exhaustive list: APP, FDR, and others.

2-       Figure 4. This reviewer believes if panels b and c are to remain blank, that they should be removed.

3-      Figure 2a raw data is listed in the attached supplemental file. Are the relevant raw data for Figure 2b-2d also going to be available, or deposited for public access?

4-      The supplemental excel file has a date, not a gene name, in cells G208, G267, G407- in the ICU vs control values tab.

5-      Figure A3 supports the authors interest in further assessment of the protein folding network. This reviewer suggests either including that into a manuscript figure, or allocating Figure 7 to the appendix as well.

6-      Per the manuscript line 312 ‘Most interestingly, we were able to find the possible marker to differentiate between decedents (ICU A) and survivals (ICU B). STAT1 was significantly more highly regulated in survivals compared to…’ Did the authors attempt a validation assay following this observation?

7-      Line 615- ‘The protein CA-1 was found …. and shows a significant increase in the deceased ICU patients.’ Can the author’s clarify whether this sentence? Figure 4 only compares ICU groups to the control, and depicts both ICU B (survivors) and NW patients as significantly higher protein than the healthy controls.

NA

Reviewer 2 Report

The study by Woolly et al is focused on an interesting and important issue of the biomarkers that can be used to improve treatment and prognosis in COVID-19. The manuscript is logically designed and well written. 

There is only one, but very substantial issue that needs to be addressed. The results are not sufficiently presented. The data set deposited to ProteomeXchange could not be accessed. These data are not presented in the paper or supplemental data to the paper in sufficient detail; some data are presented in Fig. 2, Table 2, Table A3, Table S2, but they are fragmentary and incomplete. For instance, Table S2 shows p- and q-values of the t-test for several proteins, but does not tell us anything about their levels in various patient populations. It is imperative to see the actual data generated in this study for individual proteins, so a reader could draw her/his own conclusions.
